# Chemical Composition, Antioxidant, and Antimicrobial Activities of *Vetiveria zizanioides* (L.) Nash Essential Oil Extracted by Carbon Dioxide Expanded Ethanol

**DOI:** 10.3390/molecules24101897

**Published:** 2019-05-17

**Authors:** Andreea David, Fang Wang, Xiaoming Sun, Hongna Li, Jieru Lin, Peilei Li, Gang Deng

**Affiliations:** 1Department of Biotechnology, College of Chemistry and Life Sciences, Zhejiang Normal University, Jinhua 321004, China; 201720800091@zjnu.edu.cn (A.D.); sky100@zjnu.cn (F.W.); sunxm64@zjnu.cn (X.S.); 201720200659@zjnu.edu.cn (H.L.); 201720200680@zjnu.edu.cn (J.L.); 201620200631@zjnu.edu.cn (P.L.); 2Department of Biotechnology, Xingzhi College, Zhejiang Normal University, Jinhua 321004, China

**Keywords:** vetiver essential oil, CO_2_ expanded ethanol, hydrodistillation, chemical composition, antimicrobial activity, DPPH free radical scavenging assay

## Abstract

In the present study, the composition of essential oil isolated from the roots of *Vetiveria zizanioides* (L.) Nash, harvested in China, was studied, along with the bioactivities. A green novel method using an eco-friendly solvent, CO_2_-pressurized ethanol, or carbon dioxide expanded ethanol (CXE) was employed to isolate the essential oil from the root of *Vetiveria zizanioides* (L.) Nash with the purpose of replacing the traditional method and supercritical fluid extraction (SFE). After investigating the major operating factors of CXE, the optimal conditions were obtained as follows: 8.4 MPa, 50 °C, 5 mL/min ethanol, and 0.22 mole fraction of CO_2_, presenting an extraction oil that ranged from 5.12% to 7.42%, higher than that of hydrodistillation (HD) or indirect vapor distillation (IVD). The Gas Chromatography-Mass Spectrometry (GC-MS) analysis showed that three major components, including valerenol (18.48%), valerenal (10.21%), and β-Cadinene (6.23%), are found in CXE oil, while a total of 23 components were identified, 48 components less than using conventional hydrodistillation. Furthermore, the antimicrobial activities of root oils were evaluated by the microdilution method, which showed that CXE oil exhibited an ability against Gram-positive bacteria, especially *Staphylococcus aureus*, approximately equivalent to traditional samples. Additionally, the DPPH free radical scavenging assay demonstrated that the antioxidant abilities of root oils were sorted in the descending order: IVD > HD > CXE > SFE. In conclusion, after a comprehensive comparison with the conventional methods, the CXE-related technique might be a promising green manufacturing pattern for the production of quality vetiver oil, due to the modification of ethanol by the variable addition of non-polar compressible CO_2_, ultimately resulting in a prominent dissolving capability for the extraction of vetiver solutes.

## 1. Introduction

*Vetiveria zizanioides* (L.) Nash, syn. *Chrysopogon zizanioides* (L.) Roberty, is a perennial plant of the *Poaceae* family (subfamily *Panicoideae*), native from India, and widely found in Asia, Africa, Oceania, Central, and South America [1]. The volatile oil from the vetiver root is a viscous liquid, with diverse colors from pale-yellow to dark brown, as well as a deep, smoky, earthy, and woody odor [2]. Vetiver oil is commonly used as a main odor contributor in the perfumery industry and as a flavor agent in the food industry [1,3]. Despite its special aroma, vetiver oil also possesses various biological activities, such as antioxidant [4], antibacterial [5], and anti-inflammatory properties [6], making it beneficial in aromatherapy [7]. In fact, the annual market demand of vetiver oil was estimated at up to 250 tons, worth approximately $200 million per year [8], while the worldwide herbal cultivating area has reached nearly 10,000 hectares [9]. Because of these introductory facts, many researchers have focused on the innovation of extraction techniques to reduces production costs as well as environmental pollution.

Hydrodistillation is the most widely used conventional method for the isolation of volatile compounds from plant materials [10]. In general, the traditional extraction of vetiver oil using hydrodistillation usually lasts over 10 h [11,12] at relatively high temperatures (steam or boiling water) [13], resulting in the break-down of some oxidizable and thermolabile components as well as a severe decline of the oil yield [14]. However, in recent years, many articles reported a novel extraction method for an oil-fat mixture [15,16] based on CO_2_ expanded ethanol (CXE) or CO_2_-pressurized ethanol, which has been successfully used for the isolation of carotenoid pigment accumulated in *Haematococcus pluvialis* [17], *α-*pinene, and *cis-*verbenol from tree resin (*Boswellia sacra*) [16] and phenolic compounds from garlic husk (*Allium sativum* L.) [18,19].

Apparently, although the major component of CXE is a typical polar solvent, ethanol, the addition of CO_2_ may effectively promote the inherent properties, including the specific dissolving ability as well as the mass-transfer efficiency [20,21,22], being able to extract relatively complex oil from the matrix [14]. Different from supercritical fluid (SCF) [11], CXE has an enhanced solubility for various solutes, which can be used for the extraction of both non-polar and moderately polar components [17,23]. From the above analysis, as an environmentally friendly solvent, CXE fluid may have potential in improving the fundamental and functional properties of vetiver root oil extraction.

The present study will focus on the chemical composition, antimicrobial, and antioxidant activities of the vetiver oil obtained from the novel method, CXE, in comparison with traditional methods, including hydrodistillation (HD) and supercritical fluid extraction (SFE). The current investigation aims to ascertain whether the application of a green method might improve the quality or enhance the yield of vetiver oil.

## 2. Results

### 2.1. Chemical Components of Essential Oil Extracted from Vetiveria Zizanioides Root

The chemical compositions of vetiver oils obtained from the extraction methods (CXE, SFE, HD, and IVD) are presented in Table 1. Based on the GC-MS analysis of oil samples from the four above methods, a total of 48 components were identified and then grouped into four classes, including hydrocarbons, alcohols, carbonyl compounds, and carboxylic acids. Traditional and CXE oil showed a larger variety of components, such as alcohols and carboxylic acids, with the relevant amounts of 2.77% and 2.66% detected, respectively. Also, it was found that carbonyl compounds exist in CXE oil (5.42%) and IVD oil (1.54%). Furthermore, SFE oil is mainly composed of hydrocarbons (81.91%), which can be used as a potential food additive, while HD oil has an applicable usage in the perfumery and cosmetic industry [11]. Firstly, for the root oil from traditional hydrodistillation, 21 volatiles were observed with three main chemical compounds, cedr-8-en-13-ol (26.54%), β-guaiene (15.31%), and cycloisolongifolene (11.09%), which were also identified in previous reports [24]. Meanwhile, 25 components were detected in the IVD oil, which represents an industrial sample produced by direct steam distillation. Besides the one compound, cedr-8-en-13-ol (9.74%), being the same as HD oil, cycloisolongifolene (6.56%) and khusimene (5.86%) were merely present in the IVD oil, possibly exhibiting a strong ability to inhibit lipid peroxidation according to early findings. In contrast to the HD oil, the new method of CXE extraction obtained a total of 22 components, but the main components of CXE oil, including valerenol (18.48%), valerenal (10.21%), and β-cadinene (6.23%), were distinctly different from the HD oil. Furthermore, the oil produced by another green technique, SFE, exhibited a relatively simple chemical composition, solely containing 10 components and being predominated by hydrocarbons. In addition, it should be noted that although some minor components, such as khusimene, δ-selinene, and β-vetivenene, probably accounted for a low percent rate, they might be contribute to specific odor of vetiver oil. Notably, it was certain that the variation of the chemical composition due to the use of the new method instead of conventional ones would result in a possible promotion of the profiles of vetiver oil, such as reducing the unpleasant smoky fragrance, enhancing the bioactivity, and so on.

The results presented in Table 2 show that in addition to the color, the extracting yield of vetiver oil obtained from CXE extraction is significantly different from the other three methods. The color of CXE oil was pale yellow, mostly approaching the light yellow of traditional HD oil. However, the oil yield of CXE extraction reached almost 7.42%, much larger than the 0.6% for HD, 0.3% to 0.5% for IVD, and around 0.5% for SFE, respectively. Interestingly, the industrially-produced IVD oil exhibited a dark brown color while SFE oil was represented by an unspecific color, which was brown mixed with shades of green, close to an olive color.

### 2.2. Antimicrobial Activity

The minimum inhibitory concentrations (MICs) of five different samples of vetiver essential oils against Gram-positive and Gram-negative bacteria are illustrated in Table 3. CXE oil exhibited a relatively strong antimicrobial ability against *Staphylococcus aureus* (MIC = 78 μg/mL) and a weak activity against *Bacillus subtilis* (MIC = 312.5 μg/mL), *Escherichia coli* (MIC = 312.5 μg/mL), and *Pseudomonas aeruginosa* (MIC = 2500 μg/mL). SFE oil presented a moderate activity against the same Gram positive bacteria, *Staphylococcus aureus,* similar to CXE oil, it had a poor ability against *Bacillus subtilis* (MIC = 156 μg/mL), *Pseudomonas aeruginosa* (MIC = 312.5 μg/mL), and *Escherichia coli* (MIC = 625 μg/mL). The essential oil extracted from vetiver root using HD showed a powerful antimicrobial activity against *Staphylococcus aureus* (MIC = 39 μg/mL), but also a moderate activity against *Bacillus subtilis* (MIC = 312.5 μg/mL), *Pseudomonas aeruginosa* (MIC = 312.5 μg/mL), and *Escherichia coli* (MIC = 312.5 μg/mL). The essential oil obtained from IVD, especially LFO (light fraction oil), displayed an evident antimicrobial ability against Gram-positive bacteria, *Staphylococcus aureus* (MIC = 78 μg/mL), but HFO (MIC = 156 μg/mL) displayed opposite result regarding the same bacteria. In accordance with the results, conventional and traditional methods have significant antimicrobial activity against Gram-positive bacteria, especially *Staphylococcus aureus*.

### 2.3. DPPH Free Radical Scavenging Activity of Vetiver Oils

In this study, the antioxidant activity of essential oils from *Vetiveria zizanioides* roots were evaluated by a DPPH· (1,1-diphenyl-2-picrylhydrazyl free radical) scavenging assay in contrast with a well-known hydrophilic antioxidant, ascorbic acid (Vc). The median inhibitory concentration 50 (IC_50_) values of the vetiver essential oils and the standard antioxidant, ascorbic acid, are shown in Table 4. The IC_50_ values of the five different essential oil samples ranged from 1.39 to 4.54 mg/mL. Since low values of IC_50_ represent high antioxidant activity, the ability of oil samples can be observed in the following order: IVD oil (2.19 mg/mL) > HD oil (3.57 mg/mL) > CXE oil (3.71 mg/mL) > SFE oil (4.54 mg/mL). Also shown in Figure 1, the CXE vetiver oil was anticipated to have approximately similar results with the traditional techniques.

## 3. Discussion

Since the most commonly applied method of hydrodistillation is usually performed at the laboratory scale using a Clevenger apparatus, thus representing a complete and effective extraction method for volatile compounds, HD oil might be regarded as the traditional product for vetiver root, as acknowledged by the current market. The experimental results indicated that although the number of total components of CXE oil was close to HD oil, the chemical composition exhibited a slight difference, most possibly due to the fact that CXE extraction depends on the solubility of the extracts in the specific solvent, whereas hydrodistillation obtains extracts based on the reduction of the co-boiling point while mixing and then heating the root matrix with the water. Literally, IVD, namely, indirect vapor distillation, implies that the water does not need to be in direct contact with the root matrix before distillation, and instead live steam is employed directly to penetrate the matrix for the extraction of volatile compounds, which is generally more intensive than the direct distillation (HD) performed in an ordinary laboratory [1]. Therefore, the result showed the similar chemical composition of IVD and HD, merely representing the distillation of different magnitudes. Both CXE and SFE belong to the green solvents recently recommended by many researchers, but remarkable differences between the two pressurized fluids were displayed when they were used in the extraction of natural products. Notably, for the extraction of vetiver oil from the root material, less components could be obtained by SFE, because supercritical CO_2_ might preferentially extract solutes of relatively low polarity. In other words, the polarity range of SFE might not fit that of the polarity of vetiver oil. However, different from pure supercritical CO_2_, the ethanol pressurized by compressed CO_2_ was a perfect solvent, exhibiting a more extensive polarity range, which was achieved by mixing the common polar solvent, ethanol, and the nonpolar solvent, supercritical CO_2_. [32]. As far as the polarity was concerned, CXE was more suitable to extract the vetiver root oil, whose chemical composition was more complex than that of other general plant essential oils.

Since CXE was firstly used as a solvent to produce vetiver root oil, it was necessary to find the proper extraction conditions and to optimize the operating conditions. All experiments of CXE extraction were performed at a constant temperature of 323 K, because both the previous experimental results and the analysis of literature [27] show that the temperature of CXE fluid is not significant for extraction, but should be maintained over the critical value (T_c_ = 313K). At the temperature of 323 K, the effect of three major operating parameters of CXE, including pressure, mole fraction of CO_2_, and flow rate of ethanol, on the vetiver oil yield was investigated, as shown in Figure 2. The result in Figure 2a shows that the oil yield increased significantly with the rise of pressure from 5 MPa to 9.3 MPa at a constant mole fraction of CO_2_ of 0.22 and ethanol flow rate of 5 mL/min. The pressure was a key factor that could alter the existence form of CO_2_ in the CXE from the subcritical to supercritical state, further resulting in a change of the solubility as well as other fluid properties, especially the density and viscosity, which mainly depended on the expansion extent of the systemic volume. In Figure 2b, at a constant pressure of 8.4 MPa and an ethanol flow rate of 5 mL/min, the yield of vetiver oil increased from 4.5% to 5.4% with the CO_2_ mole fraction rising from 0.16 to 0.22. The amount of CO_2_ that is added should be controlled within a relatively narrow range to ensure the intact CXE fluid exhibits a phase that is as homogenous as possible, because phasic separation is usually adverse to solute extraction. The experimental result showed that even a slight change of the mole fraction of CO_2_ might influence the polarity of the fluid, causing fluctuations of the oil yield. As illustrated in Figure 2c, the vetiver yield increased from 3.5% to 7.4% with the flow rate of ethanol changing from 3 to 8 mL/min at a constant pressure and CO_2_ mole fraction. If the extraction time was fixed for 3 h, the flow rate of ethanol mainly determined the final consumption of the CXE fluid. In particular, the slower flow rate of 5 mL/min had a higher yield of 7.42%, probably because the extracting process could be performed in adequate surroundings of the material matrix into the CXE fluid. Especially, a higher flow rate of ethanol of 8 mL/min had a lower oil yield of 5.12%, which is a result of a partial extraction of vetiver solutes due to the non-uniform excess of the solvent in the matrix. CXE exhibited a higher yield than that obtained with conventional methods (0.3%–0.6%) and SFE (0.5%).

The CXE oil exhibited a pale yellow color with a specific vetiver odor, which may be explained by the prominent dissolving capability of the extraction process. HD oil showed a similar color, light yellow, and was distinguished by a pleasant woody odor caused by the presence of khusimene (2.16%) and β-vetivenene (3.89%) [12]. On the other side, the high temperature and long time of the distillation caused IVD oil to present a dark brown color and smoky odor. Curiously, SFE exhibited a non-specific vetiver color, olive, which might be caused by the carotenoid pigments extracted from the root matrix.

The antimicrobial activity of the vetiver oil samples demonstrated a significant ability against Gram-positive bacteria, especially *Staphylococcus aureus*, as well as *Bacillus subtilis* and a relatively poor activity against the Gram negative bacilli, *Pseudomonas aeruginosa* and *Escherichia coli* [33]. Similar to the results from the literature, Gram-positive bacteria are usually more vulnerable to natural plant extracts compared to Gram-negative bacteria [2]. According to the literature [34], the strong antimicrobial activity of HD oil can be attributed to the presence of alcohols (2.77%). In addition, the moderate activity of CXE can be justified by ketones (1.54%), as it is reported that they can exhibit a potential ability.

The CXE vetiver oil presented an approximately similar antioxidant activity as traditional oil. Based on the other references [35], the presence of β-vetivenene can justify the slight ability of IVD oil (0.73%) and the strong antioxidant activity of HD oil (3.89%). The essential oil obtained by the SFE method has weak antioxidant activity because of its poor chemical composition, which was predominantly composed of hydrocarbons [34].

## 4. Materials and Methods

### 4.1. Plant Material

Fresh roots of *Vetiveria zizanioides*, over 2 years old, were harvested in December 2017 from Lanxi (29.208°N, 119.485°E, 38 m above sea level), Zhejiang, China. The commercial samples of vetiver oil, including light fraction oil (LFO) and heavy fraction oil (HFO), were supplied by a local producer, Natural Perfume Factory, Lanxi, Zhejiang province. The vetiver roots were cleaned to remove the soil, and then were air-dried at room temperature for 48 h. The dried materials were cut into small pieces (the length was less than 1 cm). Then, they were ground into a root powder in liquid nitrogen. The experimental root materials were stored at −20 °C in an ultra-low refrigerator (New Brunswick, Model Innova U725-86, UK).

### 4.2. Extraction Techniques of Essential Oil

#### 4.2.1. Carbon Dioxide Expanded Ethanol Extraction

The CXE extraction experiments were performed using a Supercritical Fluid Extraction device (HELIX Applied Separations, Model 7307, Allentown, PA, USA). In total, 25 g of weighed root powder was mixed with 6 g of glass beads (aimed at increasing the contact of the fluid with the root matrix to facilitate the isolation of the oil) and then loaded into a stainless-steel extraction vessel. The experimental design ensured a constant temperature of 323 K, and CO_2_ was pressurized by the syringe pump to the desired mole fraction of 0.22% measured at an operating pressure of 8.4 MPa. Meanwhile, the ethanol was pumped by the HPLC pump (K-501, Knauer, Berlin, Germany) at a flow rate of 5 mL/min to form the CO_2_ expanded ethanol (CXE). CXE fluid flowed into the extraction vessel at the operating temperature, followed by the separation vessel at ambient pressure, and collected by the peristaltic pump every 10 min for a total extraction time of 150 min. Solvent was removed by rotary evaporation at 50 °C and the oil samples were stored in a refrigerator for further analysis.

#### 4.2.2. Supercritical Fluid Extraction

The supercritical fluid extraction (SFE) was carried out using the same device mentioned above. About 40 g of root powder mixed with 70 g of glass beads were loaded into the extraction vessel. The process of extraction was performed under the following conditions: Constant temperature of 323 K, pressure of 20 MPa, flow rate of CO_2_ operated at 60 mL/min, and a total extraction time of 90 min.

#### 4.2.3. Hydrodistillation

Approximately 30 g of vetiver root powder was mixed with 0.2 L of pure water and distilled using a Clevenger apparatus. After 12 h of distillation, pale-yellow essential oil was dehydrated by anhydrous sodium sulphate, then weighed for calculation of the extraction yield. The oil samples were stored in amber-colored glass bottles at a low temperature.

#### 4.2.4. Indirect Vapor Distillation (Live Steam Distillation)

A factory situated in Lanxi city uses this conventional technique for the isolation of vetiver essential oil. Briefly, steam was used directly to pass through the vetiver root matrix (small pieces of around 5 cm). The oil-water mixture was obtained after condensation using a specific separator. The commercial oil products, including LFO (density of less than 1 g/mL) and HFO (density larger than 1 g/mL) were collected, respectively. The aspect of LFO is similar to the experimental oil obtained from HD. The middle phase between LFO and HFO is represented by the aromatic water (hydrolat), which can be recycled for production or extraction. The HFO was identified as a sticky, brown oil, possibly containing an abundance of plant lipids. The total yield of commercial vetiver oil was approximately 0. 6% *(w/w)*. Based on the physical properties, the LFO oil sample was investigated for further analysis.

#### 4.2.5. Estimation of Vetiver Oil Yield

The vetiver oil was periodically collected during the traditional hydrodistillation and non-conventional extractions, CXE and SFE, for determination of the oil yield as follows:(1)$$\mathrm{Yield}\;(\%)\;=\;\frac{\mathrm{weight}\;\mathrm{of}\;\mathrm{oil}\;\mathrm{extracted}(g)}{\mathrm{dry}\;\mathrm{weight}\;\mathrm{of}\;\mathrm{root}\;\mathrm{matrix}(g)}\;\times \;100$$

### 4.3. Gas Chromatographic/Mass Spectrum Analysis of the Composition of the Vetiver Oils

The vetiver essential oils from different extraction techniques were analyzed by GC-MS. The gas chromatography-mass spectrometry (GC-MS) analyses were performed using an Agilent GC 7820A (Agilent Technologies, Model 7820A G4350A, Santa Clara, CA, USA) coupled with an Agilent 5975C mass selective detector (Agilent Technologies, Model 5975C VL, USA). The MS detector was operated in the electron ionization (EI) mode (electron energy = 70 eV), scan range = 4–400 mAU, and scan rate = 3.99 scans/s). GC was equipped with a flame ionization detector (FID) and an HP-5ms capillary column (30 m × 0.25 mm, i.d × 0.10 μm, film thickness). Helium was used as a carrier gas with a column head pressure of 48.7 kPa and a flux of 1 mL/min [36]. Samples of 1 μL were injected using the split mode (split ratio 1:50). The injector and detector temperatures were 250 and 220 °C, respectively. The oven temperature was set as follows: Initial temperature of 50 °C, held for 5 min; increased at 3 °C/min to 240 °C, and maintained at 240 °C for 10 min. The chemical compounds of vetiver essential oil were identified based on the retention indices of the homologous series of *n*-alkanes, and by comparing their mass spectral with those reported in literature [1,11,12,24,27,28,29,30], as well as the NIST11 library, Wiley10 library and FFNSC1.2 library.

### 4.4. Antimicrobial Activity

Four bacterial strains were used for antimicrobial ability screening: Gram-positive bacteria, including *Staphylococcus aureus* (ATCC No. 6538) and *Bacillus subtilis* (ATCC No. 14579); Gram-negative bacteria, including *Pseudomonas aeruginosa* (ATCC No. 27853) and *Escherichia coli* (ATCC No. 8739). The antimicrobial activity against the above bacteria was evaluated by determining the minimum inhibitory concentration (MIC) using the microbroth dilution method. Each well of 96 microplates was aliquoted with 50 μL of cation-adjusted Mueller Hinton broth (CAMHB) (Qingdao Hope Bio-Technology Co., Ltd.); the wells of the first row contained 50 μL of pure dimethyl sulfoxide (DMSO) (Shanghai Aladdin Bio-Chem Technology Co., Ltd., Shanghai, China) (1st lane, negative control), 1% Gentamicin (Shanghai Aladdin Bio-Chem Technology Co., Ltd.) (2nd lane, positive antibiotic control), and 2.5 mg/mL oil samples were serially diluted (from 3rd lane to the 12th lane) and dissolved in DMSO, respectively. The following dilution was performed by transferring 50 μL in each well from the first row to the further one, until the eighth row. The final 50 μL of the solution in the last row was discarded. The tested strains at a concentration of approximately 1.5 × 10^8^ colony forming units (CFU)/mL were inoculated into each well, after 24 h of incubation at 37 °C [36]. Determination of the absorbance at 610 nm was used to detect the absence of turbidity in growth broth, which indicates the effectiveness of the microbial inhibition caused by the tested samples. The final MIC values were determined by the lowest concentrations, and all antimicrobial tests were performed in duplicate.

### 4.5. DPPH Free Radical Scavenging Activity of Vetiver Oils

The antioxidant activity of the vetiver essential oil was measured by the DPPH (1,1-diphenyl-2-picrylhydrazyl) free radical scavenging assay. The gradual concentrations of the essential oil samples, i.e., 10, 20, 40, 60, 80, and 120 mg/mL, were prepared accurately in ethanol. For each well of the 96 microplates, 0.15 mL of a 0.05 mmol/L ethanolic DPPH was mixed with the tested 0.15 mL oil sample [36]. The reaction liquid was incubated in the dark at room temperature for 30 min and the absorbance was read at 517 nm using a Multimode Microplate Reader (Tecan, Infinite M200 PRO, AU). The standard antioxidant, ascorbic acid, was used as a negative control. The free radical scavenging activity of each solution was calculated according to the following formula:(2)$$I_{\mathrm{DPPH}\cdot }\%\;=\;(1\;-\;\frac{A_{\mathrm{black}}}{A_{sample}})\;\times \;100$$

*A_blank_* was the absorbance of the negative control, along with the absorbance of the vetiver essential oil represented by *A_sample_*. The median inhibitory concentration (IC_50_) was calculated based on a linear regression equation.

## 5. Conclusions

In this paper, a novel green solvent, CXE, was first applied to extract quality root oil from *Vetiveria zizanioides* (L.) Nash. CXE fluid possessed a strong ability to maintain the more characteristic components, resulting from the varying range of dissolution caused by the regulated fraction of CO_2_ and the high proportion of ethanol. In the CXE oil, the composition 22 components were identified, including valerenol (18.48%), valerenal (10.21%), and β-cadinene (6.23%), showing an extension of the polarity range from polar to non-polar. Moreover, from the antimicrobial activity assay, CXE oil exhibited a moderate ability against Gram-positive bacteria, especially *Staphylococcus aureus*, which was approximately equivalent to traditional samples. Preliminary results of the antioxidant activity from the DPPH scavenging assay were also investigated, presented in the following descending order: IVD > HD > CXE > SFE. In summary, CXE–related extraction, an environmentally friendly technology, is an alternative for the production of vetiver root oil, which basically satisfies and, in some respects, exceeds traditional product criteria. Since the chemical composition of extracts from the root material are always more complex than typical essential oils from other vegetative parts of plants, further development of this novel technique as well as the following refining process is necessary.

## Figures and Tables

**Figure 1 molecules-24-01897-f001:**
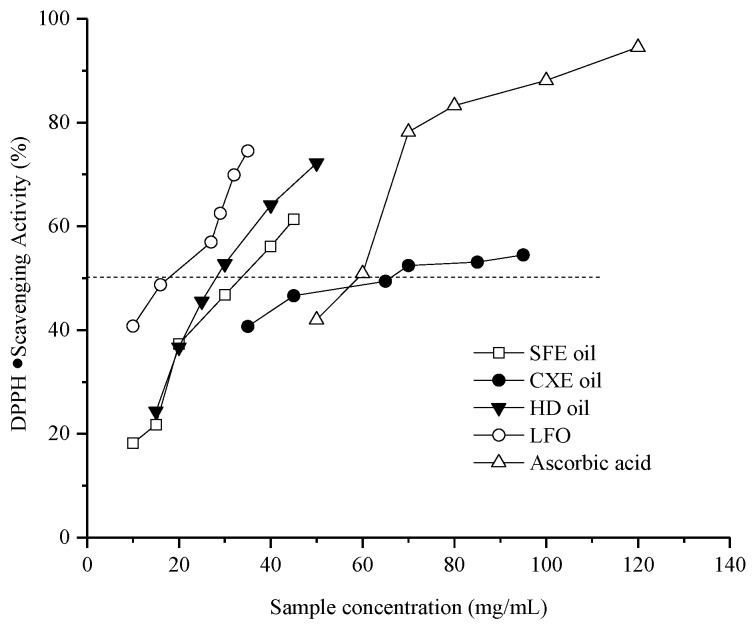
DPPH free radical scavenging ability of different samples of oil extracted from *Vetiveria zizanioides* root.

**Figure 2 molecules-24-01897-f002:**
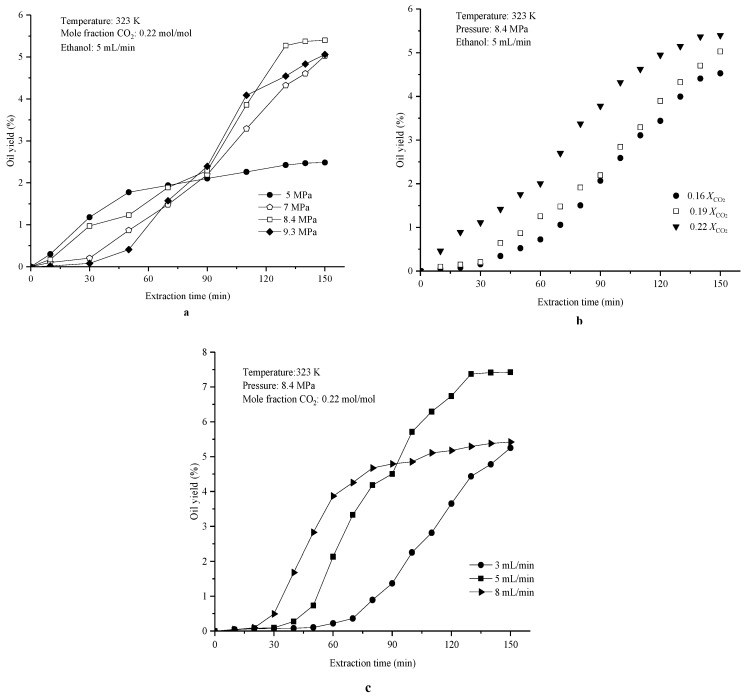
Effect of pressure, ethanol flow rate, and mole fraction of CO_2_ on vetiver oil yield at a fixed temperature of 323 K. (**a**) Effect of pressure on vetiver oil yield at a constant CO_2_ mole fraction of 0.22 and an ethanol flow rate of 5 mL/min. (**b**) Effect of CO_2_ mole fraction on vetiver oil yield at a constant pressure of 8.4 MPa and an ethanol flow rate of 5 mL/min. (**c**) Effect of the ethanol flow rate on vetiver oil yield at a constant pressure of 8.4 MPa and a CO_2_ mole fraction of 0.22.

**Table 1 molecules-24-01897-t001:** Identification of chemical components in vetiver essential oils.

No	Compounds	KI ^a^	RI ^b,c^	HD Oil ^d^ Area, %	IVD Oil ^e^ Area, %	CXE Oil ^f^ Area, %	SFE Oil ^g^ Area, %
1	cycloisolongifolene ^h^	1319	1311 [25]	11.09	6.56	4.89	
2	α-ylangene ^h^	1375	1363 [11],1371 [12],1373 [25],1465 [1]		0.64		
3	α-copaene ^h^	1376	1377 [26]	1.04	0.18	0.12	
4	isoledene ^i^	1377	1377 [25]				4.79
5	β-patchoulene ^h, j^	1381	1395 [25]		0.92	3.15	1.98
6	isolongifolene ^h^	1390	1387 [25]		0.15		
7	sativene ^h^	1391	1339 [24],1394 [25]	1.7		1.11	
8	longifolene ^h^	1408	1415 [26]			3.69	
9	α-gurjunene ^h^	1409	1419 [25]	1.15	1.02	1.38	
10	thujopsene ^h^	1431	1429 [25]			1.1	
11	β-humulene ^h^	1438	1454 [25]		0.11		
12	prezizaene ^h^	1446	1375 [1],1438 [27],1441 [11], 1449 [28],1450 [29],1459 [30]				2.25
13	khusimene ^h^	1455	1443 [27],1452 [28],1455 [25], 1462 [30],1468 [1]	2.16	5.86	1.48	4.44
14	dehydroaromadendrene ^i^	1462	1541 [25],1545 [24]			3.11	
15	β-cadinene ^i^		1472 [25]	3.05	1.05	6.23	
16	γ-muurolene ^i^	1479	1477 [26],1480 [25]		0.19		
17	α –curcumene ^h^	1480	1468 [30],1473 [12],1482 [25], 1486 [26]				11.92
18	γ-himachalene ^i^	1482	1470 [25],1476 [26]		1.87		32.65
19	α-amorphene ^i^	1484	1474 [30],1452 [25],1478 [27], 1481 [11],1491 [1]			3.54	
20	cis-eudesma-6,11-diene ^h^	1489	1478 [12],1487 [25],1490 [27]		3.73	2.46	
21	β-selinene ^h^		1485 [26],1488 [25]	0.30			
22	β-guaiene ^i^		1485 [25], 1490 [26],1523 [24]	15.31	1.02		
23	eudesma-4,6-diene (δ-selinene) + ^h^	1492	1484 [12],1490 [27],1493 [11,25]		4.93		
24	β-vetispirene ^i^	1493	1506 [1],1484 [12],1488 [25,27], 1491 [11]	2.23			
25	zinigiberene ^h^	1493	1523 [26]			0.22	
26	cadina-1,4-diene (cubenene) ^h^	1495				0.16	
27	valencene ^h^	1496	1491 [25],1522 [11],1495 [28]	0.46		0.18	
28	eudesma-3,11-diene (α-selinene) ^h^	1498	1480 [25],1492 [12]	2.18			1.36
29	α-muurolene ^h,i^	1500	1499 [25],1503 [28]		3.4		2.46
30	δ-guaiene ^h^		1508 [25]			0.38	
31	γ-cadinene ^i,j^	1513	1508 [28],1512 [12],1514 [25], 1519 [11]		0.12		18.08
32	δ –cadinene ^h^	1523	1502 [28],1508 [30],1517 [12], 1524 [26],1526 [25],1529 [11]	1.15	0.85	4.06	
33	β-vatirenene ^i^	1544	1527 [25],1554 [26]		0.64		
34	α-calacorene ^h^	1545	1533 [12],1544 [27],1546 [25], 1547 [11],1552 [1]	0.59	1.31	0.45	
35	β-vetivenene ^h^	1555	1546 [12],1552 [25,27],1556 [11], 1574 [1]	3.89	0.73		
36	zierone ^h^	1575	1754 [25]		0.87	3.48	
37	10-epi-γ-eudesmol ^h^	1623	1624 [25]	2.77			
38	longifolenaldehyde ^h^		1668 [25]		0.48		
39	alloaromadendrene epoxide ^i^	1641	1646 [25]	6.36			
40	cubenol ^i^	1646	1580 [24],1642 [25]	1.47			
41	valerenol ^h^		1655 [25]			18.48	
42	valerenal ^h^		1688 [25]	5.18	5.09	10.21	
43	cadina-4,9-diene ^h^		1670 [26]	4.86			
44	β-nootkatol ^i^		1712 [25],1722 [26]		0.67		
45	cedr-8-en-13-ol ^i^	1689	1688 [25],1769 [24]	26.54	9.74		
46	nootkatone ^h^	1809	1809 [27],1812 [11]				1.98
47	zizanoic acid ^i^	1811	1817 [27],1818 [11],1837 [1], 1860 [25,28]	2.66			
Hydrocarbons				90.71	50.59	64.46	81.91
Alcohols				2.77			
Carbonyl compounds					1.54	5.42	
Carboxylic acids				2.66			
Total identified				96.14%	52.13%	69.88%	81.91%

Note: ^a^ Kovats index (KI) on DB-5 in reference to *n*-alkanes [31]; ᵇ Retention Index on DB-5 in reference [25,31]; ^c^ References Retention Index: [11,27] Adams Library and Wiley5 library on HP-5 columns; [29] Individual library on DB-5 column; [24,12] NIST library on ID-BPX5 and BP-1columns; [28,30] NIST98, Wiley5 and Wiley275 libraries on HP-1 and HP-5 columns; [1] Data bank NBS75K on HP-1 column; ^d^ Vetiver oil obtained from conventional hydrodistillation (HD); ^e^ IVD oil supplied by Natural Perfume Factory, Lanxi, Zhejiang province; ^f^ Oil extracted from vetiver grass by carbon dioxide expanded ethanol extraction (CXE); ^g^ Vetiver oil extracted by supercritical fluid extraction (SFE); ^h^ Chemical component identified by WR10 Library; ^i^ Chemical component identified with NIST11 Library; ^j^ Chemical component identified using FFNSC1.2 Library.

**Table 2 molecules-24-01897-t002:** General comparison of vetiver essential oils obtained from four different methods.

Vetiver Essential Oil	Color	Yield (%)	Extraction Time (h)
HD oil	pale yellow	0.6	12
IVD oil	dark brown	0.3–0.5	18
CXE oil	light yellow	5.12	3
SFE oil	greenish brown or olive	0.5	2

**Table 3 molecules-24-01897-t003:** Minimal inhibitory concentration of essential oil extracted from *Vetiveria zizanioides* root.

Sample	Antimicrobial Activity (MIC, μg/mL)			
	*S. aureus* (G+) ^a^	*B. subtilis* (G+)	*P. aeruginosa* (G−) ^b^	*E. coli* (G−)
HD oil	39	312.5	312.5	312.5
IVD oil	78	2500	2500	312.5
CXE oil	78	312.5	2500	312.5
SFE oil	78	156	312.5	625

**Table 4 molecules-24-01897-t004:** Median inhibitory concentration (IC_50_) of the vetiver oils derived from linear regression.

Samples	Regression Equation	R2	IC_50_ (mg/mL)
SFE oil	*y* = 9.38*x* + 7.43	0.97	4.54
CXE oil	*y* = 2.61*x* + 40.31	0.90	3.71
HD oil	*y* = 9.39*x* + 16.426	0.99	3.57
IVD oil (LFO)	*y* = 6.80*x* + 35.09	0.99	2.19
Ascorbic acid	*y* = 10.84*x* + 34.90	0.90	1.39

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
