# Peer review of "Chemical Composition, Antioxidant, and Antimicrobial Activities of Vetiveria zizanioides (L.) Nash Essential Oil Extracted by Carbon Dioxide Expanded Ethanol"

_molecules, 2019, doi:10.3390/molecules24101897_

Round 1

Reviewer 1 Report

Reviewer's Comment:

The manuscript is interesting and describes the chemical composition, antioxidant and antimicrobial activities of Vetiveria zizanioides (L.) Nash essential oil extracted by Carbon Dioxide Expanded Ethanol.

Line 2: “Vetiveria zizanioides” must be written in italic.

Line 12-264:Nash” please check and uniform the style (not in italic).

Line 38: “(pp. 16-19)” what mean?

Line 47: “(pp. 164-167)” what mean?

Line 50: “SFE” this abbreviation is not reported in extenso. Please check.

Line 56: “IVD” this abbreviation is not reported in extenso. Please check.

Line 92: Table 2 “FLO” this abbreviation is not reported in extenso. Please check.

Line 103: “FLO” this abbreviation is not reported in extenso. Please check.

Lines 147, 151 and 156: please check the use of bold style.

Lines 169-170: “The antimicrobial activity of vetiver oil samples have a significant effectiveness against Gram-positive bacteria, especially Staphylococcus aureus and a low activity against Gram-negative bacilli.” Considering that you have tested only one for each bacteria is not possible to extend the results to all these bacteria. Is reported in fact, in several paper such as (Evaluation of the antibacterial activity of bergamot essential oils on different Listeria monocytogenes strains. 2016 Italian Journal of Food Safety Volume 5, Issue 4, 8p) that bacteria are characterized by a strong variability in the response to the essential oils. Please limit the results to the tested bacteria.

Line 205: “…above. 40 g of…” usually in English is not correct start a sentence with number

Lines 290-372: please check and verify the reference style according to the journal requirement.

Lines 243-244: “cation-adjusted Mueller Hinton broth”, “DMSO” and “Gentamicin”, please insert the name and the producer of all the media used.

Author Response

Response to the 1st reviewer’s comments

The manuscript is interesting and describes the chemical composition, antioxidant and antimicrobial activities of Vetiveria zizanioides (L.) Nash essential oil extracted by Carbon Dioxide Expanded Ethanol.

Line 2: “Vetiveria zizanioides” must be written in italic.

Reply: We have accepted the suggestion and revised the font-style of the title.

Line 12-264: “Nash” please check and uniform the style (not in italic).

Reply: Thanks for the valuable suggestion. The style of the scientific name of the plant has been modified (line 11-265).

Line 38: “(pp. 16-19)” what mean?

Reply: Thank you for the comment about the “(pp. 16-19)” which is indicating the pages of the thesis- reference 13. As it mention in the MDPI- Instructions for Authors: “In the text, reference numbers should be placed in square brackets [ ], and placed before the punctuation; for example [1], [1–3] or [1,3]. For embedded citations in the text with pagination, use both parentheses and brackets to indicate the reference number and page numbers; for example [5] (p. 10). or [6] (pp. 101–105).” We have reconsidered and removed the brackets of the line 37.

Line 47: “(pp. 164-167)” what mean?

Reply: We have accepted the suggestion and removed the brackets “(pp. 164-167)” of the line 46. According to the MDPI- Instructions for Authors: “In the text, reference numbers should be placed in square brackets [ ], and placed before the punctuation; for example [1], [1–3] or [1,3]. For embedded citations in the text with pagination, use both parentheses and brackets to indicate the reference number and page numbers; for example [5] (p. 10). or [6] (pp. 101–105).”

Line 50: “SFE” this abbreviation is not reported in extenso. Please check.

Reply:Thanks for your valuable suggestion. “SFE” represents the abbreviation for Supercritical Fluid Extraction and it was mentioned in the abstract (line 13).

Line 56: “IVD” this abbreviation is not reported in extenso. Please check.

Reply:IVD represents the abbreviation for Indirect Vapor Distillation which was mentioned in the abstract (line 16).

Line 92: Table 2 “FLO” this abbreviation is not reported in extenso. Please check.

Reply:We accepted the suggestion and revised the table 2 (line 99). The abbreviation “LFO” represents the light fraction of industial oil which is particularly mentioned in the section 4.1 (line 198) and section 4.2 (line 223-231).

Line 103: “FLO” this abbreviation is not reported in extenso. Please check.

Reply:According to this suggestion, we revised the abbreviation through the addition of the full term light fraction oil (line 111).

Lines 147, 151 and 156: please check the use of bold style.

Reply:Thanks for your valuable suggestions. We considered that the bold style could highlight the different efects of the extraction process represented in the figure 2a-c. The bold style was removed, shown in lines 157, 160 and 165.

Lines 169-170: “The antimicrobial activity of vetiver oil samples have a significant effectiveness against Gram-positive bacteria, especially Staphylococcus aureus and a low activity against Gram-negative bacilli.” Considering that you have tested only one for each bacteria is not possible to extend the results to all these bacteria. Is reported in fact, in several paper such as (Evaluation of the antibacterial activity of bergamot essential oils on different Listeria monocytogenes strains. 2016 Italian Journal of Food Safety Volume 5, Issue 4, 8p) that bacteria are characterized by a strong variability in the response to the essential oils. Please limit the results to the tested bacteria.

Reply:We accepted the suggestion and revised line the 178-180. The detailed ability of the vetiver essential oils was described in the section 2.2. According to the reference [1], the antimicrobial activity of vetiver oil samples have a significant ability against Gram-positive bacteria, especially Staphylococcus aureus, as well as Bacillus subtilis and a relatively poor activity against Gram negative bacilli, Pseudomonas aeruginosa and Escherichia coli.

Line 205: “…above. 40 g of…” usually in English is not correct start a sentence with number

Reply:We have accepted the suggestion and revised the section 4.2 (line 215).

Lines 290-372: please check and verify the reference style according to the journal requirement.

Reply:Thanks. The reference style was revised according to the journal requirement.

Lines 243-244: “cation-adjusted Mueller Hinton broth”, “DMSO” and “Gentamicin”, please insert the name and the producer of all the media used.

Reply:We have accepted the suggestion and revised the section 4.4 (line 253-256). “Cation-adjusted Mueller Hinton broth” (CAMHB) (Qingdao Hope Bio-Technology, Co., Ltd.), dimethyl sulfoxide (DMSO) (Shanghai Alladin Biochemical Technology Co., Ltd.) and “Gentamicin” (Shanghai Alladin Biochemical Technology Co., Ltd.).

Reviewer 2 Report

The study by David and collective is interested mainly with carbon dioxide pressurised ethanol extraction of vetiveria grass. The extraction method is proposed for the industrial extraction of the vetiveria oil.  

The study itself comparison several methods of the vetiveria oil extraction. The study although quite comprehensive deals with niche subject more interesting for process engineers and industry in general. Therefore it's more suited  for more specialised  journal.

In table 1 the identification of compounds is questionable as it is based on the databases  and not on the standard chromatographic methods like co chromatography of standards. Indeed the opponent is doubtful about some of the identified components,  like number 48 ( betulin) which cannot be  as far as the opponent knows gas chromatographed without derivatization. Please explain.

As none of the co-authors is botanist, it is customary in papers dealing with the plant material To get formal identification by the Department of Botany  or analogical Institute and voucher  plant specimen is to be stored  in botanical collection for possible further investigation.

The reference section is to be amended and checked properly. Clearly the reference manager used sometimes messed the format of the citation,  like number 31. Also reference the identified compounds with proper reference of the paper where the compound was first identified. Do not use general reference to database only.

Author Response

Response to the 2nd reviewer's comments:

The study by David and collective is interested mainly with carbon dioxide pressurised ethanol extraction of vetiveria grass. The extraction method is proposed for the industrial extraction of the vetiveria oil.

The study itself comparison several methods of the vetiveria oil extraction. The study although quite comprehensive deals with niche subject more interesting for process engineers and industry in general. Therefore it's more suited  for more specialised  journal.

In table 1 the identification of compounds is questionable as it is based on the databases  and not on the standard chromatographic methods like co chromatography of standards. Indeed the opponent is doubtful about some of the identified components, like number 48 (betulin) which cannot be as far as the opponent knows gas chromatographed without derivatization. Please explain.

Reply: Thanks for the valuable suggestion. According to this suggestion, the component number 48 (betulin) was removed, shown in Table 1. As mentioned in the reference [1], the identification of essential oil compounds is inaccurate because of the overlapped peaks and also of components with the same GC retention time which due to the chemical complexity of vetiver oil. In addition, although the recent method to identify the composition of plant essential oils is mostly based on databases, our further study about refining the vetiver oil using molecular distillation (MD) is going to consider using the authentic standards to make precise identification of chemical composition.

As none of the co-authors is botanist, it is customary in papers dealing with the plant material To get formal identification by the Department of Botany or analogical Institute and voucher plant specimen is to be stored  in botanical collection for possible further investigation.

Reply: Thanks for your suggestions. In this article, the experimental material, roots of Vetiveria zizanioides was supplied by a local producer, Natural Perfume Factory, Lanxi, Zhejiang province, according to the experiment-designing requiements. Also, the plant material had been identified by the experts in the department of botany in our university.

The reference section is to be amended and checked properly. Clearly the reference manager used sometimes messed the format of the citation, like number 31. Also reference the identified compounds with proper reference of the paper where the compound was first identified. Do not use general reference to database only.

Reply: We accepted this suggestion and modified the reference style. We used the database and general reference, but not only also we checked in the reference [2] for a complete identification of the compound.

Reviewer 3 Report

Evaluation of the paper entitled “Chemical composition, antioxidant and antimicrobial, activities of Vetiveria zizanioides (L.) Nash essential oil extracted by Carbon Dioxide Expanded Ethanol” by Deng et al.

General comments: Please verify the guide authors for the authors' names and references. They do not follow the journal’s rules.

Abstract

I recommend the authors to begin the abstract with a couple of introduction lines, in order to highlight the importance of the essential oil chosen.

Line 16: By CG-MS is not correct, it should be: The CG-MS analysis showed…

Line 16: remove “including” because the sentence finish as: were found…and does not look right.

I strongly recommend reviewing the English grammar of the abstract, because several lines need correction.

At the end is not totally clear what is the biggest improvement with the CXE method as compared to traditional methods, because they seem similar in properties. The authors should include a more consistent and stronger conclusion with the methodology presented that demonstrates this method as a very promising methodology to isolate essential oils.

Keywords: normally five words are presented. Reduce the number of keywords.

Introduction

Line 28: native to India must be corrected to native from India

Line 52: add the word at the end “extraction”.

In general, in the introduction section is necessary to review in several lines, to add “coma” before the word “and” when the authors are making a list of several things. Please check and change in the whole section.

Results

Line 57: On basis should be changed by based on the…

Lines 57 and 58 needs a review of the grammar and be rewritten.

Line 64: Found in previous reports instead of literatures.

Line 69: Oil produced instead of produce.

Line 72: “All in all” expression is not used frequently, please look for an equivalent expression.

Please, carefully check the footnote of Table 1, there are several grammar mistakes.

I suggest presenting first the paragraph that explains Table 2 and after that, to introduce it. It is always better to mention the table right before present it.

Table 2 should also have the refractive index and density of the different oils if it is going to be called “Physical properties of the oils”

From the antibacterial activity, if authors are making comparisons between the methods, it should be mentioned which are the compounds that are influencing the strong antibacterial activity and why. It is always important to support these comparisons to conclude that the actual method is better to produce essential oils for this application.

I suggest evaluating antioxidant activity for more than one methodology. DPPH is only a radical scavenger but it does not represent a broad range of antioxidant compounds and subsection 2.3 could not be called as “Antioxidant activity”. It is necessary at least three or four different methods to be called like that, with different kind of antioxidant mechanisms such as radical scavengers, FRAP, ORAC, and so on.

Lines 114 and 115 if the authors try to organize from lower to higher antioxidant capacities, I think the order should be inverted.

Discussion

It is very important to increase the scale of the figures to be able to read it. I am not sure if the authors claimed that they “optimize” conditions, which was the statistic support to claim that. It is necessary to shoe an experimental design and statistics support to claim an optimization process.

Materials

DPPH is a known procedure. Please, cite the reference for this method.

In general, it is necessary to review the English grammar of the whole manuscript. Please, highlight the advantage of this method in front of the other known methods for this research, using your results in the conclusion section.

Author Response

Response to the 3rd reviewer's comments:

Comments and Suggestions for Authors

Evaluation of the paper entitled “Chemical composition, antioxidant and antimicrobial, activities of Vetiveria zizanioides (L.) Nash essential oil extracted by Carbon Dioxide Expanded Ethanol” by Deng et al.

General comments:

Please verify the guide authors for the authors' names and references. They do not follow the journal’s rules.

Reply: We accepted this suggestion and had carefully modified the authors' names and reference style according to the journal requirement.

Abstract

I recommend the authors to begin the abstract with a couple of introduction lines, in order to highlight the importance of the essential oil chosen.

Reply: We have accepted the suggestion and revised the abstract (line 10-11).

 Line 16: By CG-MS is not correct, it should be: The CG-MS analysis showed…

Reply: According to this suggestion, we revised the line 16.

Line 16: remove “including” because the sentence finish as: were found…and does not look right.

Reply:We accepted this suggestion and removed “including” (line 16).

I strongly recommend reviewing the English grammar of the abstract, because several lines need correction.

Reply: We revised the English grammar of the abstract.

At the end is not totally clear what is the biggest improvement with the CXE method as compared to traditional methods, because they seem similar in properties. The authors should include a more consistent and stronger conclusion with the methodology presented that demonstrates this method as a very promising methodology to isolate essential oils.

Reply: Thank you for the comment about the conclusion of the abstract.We have reconsidered and modified the last sentence of the abstract.

Keywords: normally five words are presented. Reduce the number of keywords.

Reply: We have removed the following key words: “vetiver essential oil” and “supercritical fluid extraction”.

Introduction

Line 28: native to India must be corrected to native from India

Reply: We accepted the suggestion and revised the line 32.

Line 52: add the word at the end “extraction”.

Reply: Thanks for your suggestion. We revised the following line: “From the above analysis, as an environmentally friendly solvent, CXE fluid may have potential in improving the fundamental and functional properties of vetiver root oil extraction.”

In general, in the introduction section is necessary to review in several lines, to add “coma” before the word “and” when the authors are making a list of several things. Please check and change in the whole section.

Reply: We have accepted the suggestion and revised the introduction section.

Results

Line 57: On basis should be changed by based on the…

Reply: We have reconsidered and modified the sentence (line 61).

Lines 57 and 58 needs a review of the grammar and be rewritten.

Reply: We accepted this suggestion and modified the line 60-64.

Line 64: Found in previous reports instead of literatures.

Reply:Thanks for your comment. We have replaced the word (line 69).

Line 69: Oil produced instead of produce.

Reply: Thank you for the comment. We have revised the line 75.

Line 72: “All in all” expression is not used frequently, please look for an equivalent expression.

Reply: We have reconsidered and replaced the expression with the word “notably” ( line 78).

Please, carefully check the footnote of Table 1, there are several grammar mistakes.

Reply:Thanks. The footnote of Table 1 has been revised according to the suggestion.

I suggest presenting first the paragraph that explains Table 2 and after that, to introduce it. It is always better to mention the table right before present it.

Reply: We have accepted the suggestion and revised the introduction of Table 2 (line 92-97).

Table 2 should also have the refractive index and density of the different oils if it is going to be called “Physical properties of the oils”

Reply: According to this suggestion, the title of Table 2 of was modified.

From the antibacterial activity, if authors are making comparisons between the methods, it should be mentioned which are the compounds that are influencing the strong antibacterial activity and why. It is always important to support these comparisons to conclude that the actual method is better to produce essential oils for this application.

Reply: Thanks for the valuable suggestion. In this article, the antimicrobial ability vetiver oil extracted by Carbon dioxide expanded ethanol is compared with the traditional distillation. According to the literature [1], the strong antimicrobial activity of HD oil can be attributed to the presence of alcohols (2.77%). In addition, the moderate activity of CXE can be justified by ketones (1.54%) reported that can exhibit a potential ability. This is the first research focused on the vetiver oil extracted by CXE extraction process which is difficult to make a conclusion based on the chemical composition for explaining the antibacterial activity. In the further study, we will deeply investigate the antibacterial, as well as antifungal activity for detailed explanation.

I suggest evaluating antioxidant activity for more than one methodology. DPPH is only a radical scavenger but it does not represent a broad range of antioxidant compounds and subsection 2.3 could not be called as “Antioxidant activity”. It is necessary at least three or four different methods to be called like that, with different kind of antioxidant mechanisms such as radical scavengers, FRAP, ORAC, and so on.

Reply: We have accepted the suggestion and revised the title of the subsection 2.3, called “DPPH free radical scavenging activity of vetiver oils” (line 117).

Lines 114 and 115 if the authors try to organize from lower to higher antioxidant capacities, I think the order should be inverted.

Reply: Thanks for the comment. We have organized the antioxidant capacities from lower to higher (line 123- 24).

Discussion

It is very important to increase the scale of the figures to be able to read it. I am not sure if the authors claimed that they “optimize” conditions, which was the statistic support to claim that. It is necessary to shoe an experimental design and statistics support to claim an optimization process.

Reply: Thanks for your comments. We have increased the scale of the figures and we have not “optimized” the CXE extraction conditions. Our research purpose was to find the proper extracting conditions of the novel CXE process to obtain a high vetiver oil yield (5.12% - 7.42%) in a short extraction time (150 min) with an less impact on the environment (environmentally friendly).

Materials

DPPH is a known procedure. Please, cite the reference for this method.

Reply:We have cited the reference of the DPPH method (line 266).

In general, it is necessary to review the English grammar of the whole manuscript. Please, highlight the advantage of this method in front of the other known methods for this research, using your results in the conclusion section.

Reply: Thank you for the comment about the conclusion of this article. We have reconsidered and modified the section.

Round 2

Reviewer 3 Report

All the suggestions were addressed by the authors. Only minor spell check is required.